# Advancing the Rose Rosette Virus Minireplicon and Encapsidation System by Incorporating GFP, Mutations, and the CMV 2b Silencing Suppressor

**DOI:** 10.3390/v14040836

**Published:** 2022-04-17

**Authors:** Cesar D. Urrutia, Gustavo Romay, Brian D. Shaw, Jeanmarie Verchot

**Affiliations:** Department of Plant Pathology and Microbiology, Texas A&M University, College Station, TX 77845, USA; curru001@tamu.edu (C.D.U.); gromay@gmail.com (G.R.); bdshaw@ag.tamu.edu (B.D.S.)

**Keywords:** plant virus, bunyavirus, virion assembly, virus replication, envelope glycoproteins, minireplicon, negative strand RNA virus, reverse genetic systems, emaravirus, rose rosette virus

## Abstract

Plant infecting emaraviruses have segmented negative strand RNA genomes and little is known about their infection cycles due to the lack of molecular tools for reverse genetic studies. Therefore, we innovated a rose rosette virus (RRV) minireplicon containing the green fluorescent protein (GFP) gene to study the molecular requirements for virus replication and encapsidation. Sequence comparisons among RRV isolates and structural modeling of the RNA dependent RNA polymerase (RdRp) and nucleocapsid (N) revealed three natural mutations of the type species isolate that we reverted to the common species sequences: (a) twenty-one amino acid truncations near the endonuclease domain (named delA), (b) five amino acid substitutions near the putative viral RNA binding loop (subT), and (c) four amino acid substitutions in N (NISE). The delA and subT in the RdRp influenced the levels of GFP, gRNA, and agRNA at 3 but not 5 days post inoculation (dpi), suggesting these sequences are essential for initiating RNA synthesis and replication. The NISE mutation led to sustained GFP, gRNA, and agRNA at 3 and 5 dpi indicating that the N supports continuous replication and GFP expression. Next, we showed that the cucumber mosaic virus (CMV strain FNY) 2b singularly enhanced GFP expression and RRV replication. Including agRNA2 with the RRV replicon produced observable virions. In this study we developed a robust reverse genetic system for investigations into RRV replication and virion assembly that could be a model for other emaravirus species.

## 1. Introduction

*Fimoviridae* represents one of three genera of plant-infecting viruses with segmented negative-strand RNA (NSR) genomes within the vast order of *Bunyavirales*. The genus *Emaravirus* was established in 2012 and named for the founding member, *European mountain ash ringspot-associated virus* (EMARaV) as the sole member of the family *Fimoviridae*. Since 2012 the genus *Emaravirus* has grown to include more than 25 species. These emerging viruses are primarily devastating to trees, herbaceous woody plants, and vines in natural environments, landscapes, and agriculture. Such threatening species include EMARaV, *Rose rosette virus* (RRV), *Fig mosaic virus* (FMV), *Actinidia chlorotic ringspot-associated virus* (ACCRaV) of kiwifruit, and *Pigeon pea sterility mosaic virus-1* (PPSMV-1) [1,2,3,4]. Only two species infect monocots, *High plains wheat mosaic virus* (HPWMoV) and *Ti-ringspot associated virus* (TiRSaV) [5,6]. Moreover, new emaraviruses were uncovered in indigenous and endemic plant species. Because of the plant specialization to a particular habitat, certain emaraviruses are threatening sensitive ecosystems. Examples include the *Palo verde broom virus* (PVBV) infecting palo verde trees in the Sonoran Desert of the southwestern United States and northwestern Mexico, or the *Karaka Ōkahu purepure virus* (KŌPV) infecting the karaka tree in New Zealand [7,8].

The order *Bunyavirales,* established in 2017, has grown from nine to twelve families with negative-sense or ambisense RNA genomes [4,9,10,11]. The *Bunyavirales* include species that are among the most life-threatening diseases in humans, such as *Bunyamwera virus* (BUNV), *Crimean-Congo hemorrhagic fever virus* (CCHFV), *Hantaan virus* (HTNV), and *Rift valley fever virus* (RVFV) [12]. Recently, high throughput sequencing efforts by researchers globally have led to the identification of new tentative plant virus species across *Bunyavirales*, and therefore, the ecological or agronomic impact of these emerging viruses have not yet been fully understood. Moreover, the uptick in disease incidence for emaraviruses has created a need to investigate the mechanisms of virus replication, movement, and virus-host interactions and to develop disease-combatting strategies.

RRV causes a devastating rosette disease in cultivated roses, which is one of the most important ornamental species worldwide [2]. Cultivated roses are among the most economically important ornamental plants and the global industry is valued at an estimated USD 24 billion [13]. RRV consists of seven genome segments named RNA1 through RNA7. RNA1 encodes the viral RNA-dependent RNA polymerase (RdRp), RNA2 encodes the precursor glycoprotein (pre-GP), RNA3 encodes the nucleocapsid (N), and RNA4 encodes the putative movement protein [14]. The protein products of RRV RNA5 through RNA7 have not been characterized, although the HPWMoV RNA7 and RNA8 are suggested to be silencing suppressor proteins [15].

The first minireplicons and infectious clones developed for plant infecting nonsegmented NSR viruses were *Sonchus yellow net virus* (SYNV) for nucleorhabdoviruses and *Barley yellow striate mosaic virus* (BYSMV) for cytorhabdovirus [16,17,18]. Recently, reverse genetic systems have been reported for the plant segmented *Tomato spotted wilt virus* (TSWV; a tospovirus) and *Rice stripe virus* (RSV; a tenuivirus) [19,20]. These existing minireplicon systems use low-copy binary plasmids encoding hairpin ribozyme sequences located at the 5′ and 3′ ends of the viral antigenomic sense minireplicon RNA to minimize aberrations that may impede replication from the progeny RNAs. These systems also incorporate three or four viral suppressors of RNA silencing to suppress host defenses which are crucial platform components for initiating and maintaining virus replication.

Replicons established within *Bunyavirales* have all shown that the RdRp and N are essential and sufficient for replication [21,22,23,24]. We recently reported the first infectious clone of an emaravirus by constructing a binary vector containing a cloned cDNA copy of the RRV genome in the antigenomic orientation [25]. Infectivity was confirmed in *Rosa multiflora* and *Nicotiana benthamiana* by detection of double-stranded (ds) RNA and virions in systemic tissue. This study used the fluorescent protein iLOV as a reporter, but its fluorescence was weak and could only be observed using fluorescence/confocal microscopy [25]. Here we report the development of a new generation of RRV minireplicon consisting of plasmids encoding the viral RNA1 (RdRp), RNA3 (N), and a version of RNA5 in which the enhanced green fluorescence protein (GFP) was introduced as a visual reporter for gene expression. By including a plasmid encoding RNA2 (precursor glycoprotein; pre-GP) we observed virions by transmission electron microscopy. This innovation expands the molecular toolbox for investigating RRV replication and encapsidation.

## 2. Materials and Methods

### 2.1. Plasmids, Escherichia coli, and Agrobacterium tumefaciens Strains, and Plants

The plasmids pCB301-agR1 and -agR3 were described previously. All plasmids were maintained in *E. coli* DH5α [25]. The pCB301-agR5GFP contains the eGFP coding sequence which was PCR amplified using the R5GFPF-IF and R5GFPR-IF primers (Appendix A) and inserted into pCB301-agR5 by INFUSION^TM^ cloning (Takara Bio USA Inc., Mountain View, CA, USA) using R5F and R5R primers (Appendix A). Binary plasmids pBA002 containing the CMV 2b, TBSV p19, and the TGMV AL2 were obtained from Xiuren Zhang (Texas A&M University) [26,27,28]. All binary constructs were maintained in *A. tumefaciens* strain GV3101. The pGEMT-GFP plasmid was prepared by PCR amplification of a fragment of the eGFP sequence (extending from nucleotide positions 209–414) using GFP209F and GFP414R primers (Appendix A). *N. benthamiana* plants were grown at 23 °C with 8 h light/16 h dark.

### 2.2. GFP Fluorescence Imaging and Statistical Analysis

Agro-infiltrated *N. benthamiana* leaves were monitored using a UV lamp that is attached to fixed height accordion arm and imaged using a Nikon D3400 camera. Microscopic images were acquired on an Olympus FV3000 inverted laser scanning confocal microscope. A 20× UPLASAPO lens (O.75 NA) interfaced with Olympus Fluoview software (FV31S-SW) and 2× zoom imaged through a High Sensitivity GaAsP PMT detector at 430. In addition, 20 mg of infiltrated leaves were ground 1:5 (*w/v*) in 50 mM Tris-HCl (pH 8.0) and the soluble fraction was diluted ten-fold for fluorometric analysis using a SpectraMax iD5^®^ microplate reader (Molecular Devices, San Jose, CA, USA). We calculated the average fluorescence values (FV) per mg of fresh weight tissue (*n* = 6) and the average FV/mg of tissue treated with only R5GFP were subtracted from each of the other samples. Three experimental replicates were performed, and samples were statistically compared at each time point through ANOVA followed by Tukey– Kramer HSD posthoc test (Figure 1D). Tukey’s HSD test was performed (α = 0.05) using JMP^®^ v. 16 (SAS Institute, Cary, NC, USA).

### 2.3. Phylogenetic and Protein Structure Analysis

We recovered annotated RRV sequences from the NCBI database in Geneious Prime 2021 (Geneious, San Diego, CA, USA) to align using MUSCLE in Geneious and changes were manually identified. The i-Tasser structure and function prediction tool (Zhang lab) was used for structural modeling. PyMol (v1.7.4) (https://pymol.org/2/; accessed on 30 July 2021) was used to visualize the chosen models [29,30].

### 2.4. In Vitro Transcript Synthesis

Two µg of pGEMT-GFP plasmid was linearized using N*de*I or N*co*I, agarose gel purified, and used for in vitro transcription with SP6 or T7 RNA polymerase (New England Biolabs Inc., Ipswich, MA, USA). The reactions were terminated by adding RQ1 DNase (Promega Corp., Madison, WI, USA) and synthesized transcripts were purified using the RNA Clean and Concentrate kit (Zymo Research, Irvine, CA, USA).

### 2.5. RT-PCR and RT-qPCR of Replicon RNA from N. benthamiana Leaves

Total RNA was extracted from agroinfiltrated leaves using the Maxwell 16 LEV SimplyRNA Tissue kit (Promega Corp.). Ten micrograms of total RNA were treated with DNase I, TURBO DNA-*free* kit. cDNA was synthesized using Maxima reverse transcriptase (ThermoFisher Corp., Waltham, MA, USA), tagged gene-specific primers (final concentration of 10 nM), or random hexamers for actin controls (final concentration of 250 nM). Endpoint PCR was conducted using GoTaq polymerase (Promega Corp.), primers with 0.2 µM or 0.4 µM final concentration for viral targets or actin, respectively, and 2 µL of cDNA as template. Quantitative PCR was carried out using PowerUp^TM^ SYBR^TM^ Green Master Mix. Relative quantities were determined by using ddCt analysis with a 95% confidence interval (Appendix A).

### 2.6. Electron Microscopy

Immunosorbent electron microscopy was carried out using anti-His sera diluted 1:1000 or 1:5000 (Cytiva Life Sciences, Marlborough, MA, USA) according to [31]. *N. benthamiana* leaf punches were ground 1:10 (*w/v*) in 0.1 mM phosphate buffer (pH 6.4).

## 3. Results

### 3.1. GFP Expressing RRV Minireplicon in N. benthamiana Leaves

To create a minireplicon system suitable for reverse genetic analysis of virus replication, we employed binary plasmids encoding the antigenomic (ag) cDNAs for RNA1, RNA3, and a version of RNA5 in which GFP was introduced by replacing the ORF5 sequence (Figure 1A). These plasmids will be referred to as R1, R3, and R5GFP for simplicity. *N. benthamiana* leaves were agro-infiltrated with the mixture of cultures harboring the minireplicon, and fluorescence was visible to the naked eye with a hand-held UV lamp at 3 dpi (Figure 1B). For controls, we infiltrated leaves with a mixture of R1 plus R5GFP, R3 plus R5GFP, R5GFP alone, or the empty plasmid pCB301. The minireplicon’s fluorescence was distinctly brighter than the background seen in control leaves. Using confocal microscopy, the GFP fluorescence was evident in the cytoplasm and nucleus of epidermal cells expressing the minireplicon (Figure 1C). No fluorescence occurred in cells treated with R5GFP, or empty plasmid alone, but a low level of fluorescence appeared in cells treated with a mixture of R1 plus R5GFP and R3 plus R5GFP (Figure 1C).

At 3, 5, and 7 dpi, leaf extracts expressing the minireplicon (R1, R3, and R5GFP) produced averages of approximately 2500, 1450, and 630 FV/mg-tissue, respectively (Figure 1D). These fluorescence levels were significantly higher than those of the controls (*p* < 0.05). The RRV minireplicon consisting of R1, R3, and R5GFP produced green fluorescence that was visually and quantifiably greater than R1 or R3 alone with R5GFP. However, the pattern of declining fluorescence at 5 and 7 dpi indicated that further steps were necessary to optimize the production of progeny virus and GFP expression from the minireplicon.

### 3.2. Varying Inoculum Concentration and Introducing the p19 Silencing Suppressor to Enhance GFP Fluorescence

Segmented NSR replicon systems of animals or plants were reported in a few studies to show more robust expression after optimizing the concentration of the inoculum’s components [24,32]. We delivered reduced concentrations of *Agrobacterium* inoculum to *N. benthamiana* leaves and examined GFP expression by the RRV minireplicon using a hand-held UV lamp. First, 1.0, 0.7, and 0.3 OD_600_ concentrations of R1 were each co-delivered with 1.0 OD_600_ of R3 and R5GFP (Figure 2A). At 3 dpi the levels of GFP appeared brighter for 0.3 OD_600_ than 0.7 or 1.0 OD_600_ of R1 inoculum concentrations. In some inoculums, we co-delivered the tomato bushy stunt virus (TBSV) silencing suppressor p19 using 1.0 OD_600_ concentration which also led to somewhat higher visible fluorescence (Figure 2A). Next, leaf extracts were taken from infiltrated regions of six leaves and subjected to fluorometric analysis at 3 and 5 dpi. Three experimental replicates were performed, and the averages (FV/mg- tissue) were calculated and statistically compared within each experiment. At 3 dpi, the average values ranged from 2308 to 2880 FV/mg with the highest values reported for the 0.3 OD_600_ R1 inoculum (Figure 2B). These values declined at 5 dpi to a range of 912 to 1001 FV/mg. The values at 3 dpi and 5 dpi were not dissimilar statistically (*p* < 0.05). When p19 was included, the values at 3 dpi ranged from 1818 to 4569, with 0.3 OD_600_ of R1 inoculum providing the highest fluorescence levels that were also distinctly greater than 1.0 OD_600_ of R1 plus p19 (*p* < 0.05). At 5 dpi the values for samples containing p19 ranged between 1283 and 1854 FV/mg tissue (Figure 2B; *p* < 0.05). Although the addition of p19 led to some improvement of the fluorescence level, the values were lower at 5 dpi.

Next, we delivered 0.3 or 0.7 OD_600_ of R3 while maintaining a standard level of 0.3 OD_600_ of R1, along with 1.0 OD_600_ of R5GFP (Figure 2C). Visually the combination of 0.3 OD_600_ of R1 and R3 was higher than the 0.3 OD_600_ of R1 plus 0.7 OD_600_ of R3. At 3 dpi, the values using 0.3 or 0.7 OD_600_ of R3 were 3109 and 3083 FV/mg, respectively. At 5 dpi, the values using 0.3 or 0.7 OD_600_ of R3 were 1100 or 1410, respectively. Once again, adding 1.0 OD_600_ of p19 boosted the fluorescence values for the 0.3 or 0.7 OD_600_ of R3-containing inoculum at 3 dpi to 4622 or 4189, respectively, and at 5 dpi to 1729 or 2106, respectively (Figure 2D; *n* = 6, *p* < 0.05). These combined data suggest that including the p19 silencing suppressor and reducing the inoculum concentrations improves GFP levels. However, the decline in fluorescence at later time points showed further optimization of the replicon system was necessary for sustainable expression over time.

### 3.3. Mutations Introduced into the agRNA1 and agRNA3 Sequences Improve Replication

Since the construction of the first RRV infectious clone and replicon using the NCBI reference sequence [25], approximately 100 new RRV sequences have been reported in NCBI. We aligned the RdRp nucleotide (nt) and protein sequences of the reference RRV isolate against nine other RRV isolates using MUSCLE (Appendix A). The reference sequence for RRV RNA1 (NC_015298) had two major changes that were not present in all other reported sequences. The first change we identified was an inserted adenine at position 53 of the alignment which is located seven nts after the primary start codon of the ag strand. This frameshift mutation causes a TGA translational stop codon to occur at position 74 and the next translation start codon is located at position 107 (Appendix A). This single nt causes a 21 amino acid truncation of the N-terminal region of the viral RdRp (YP_004327589) and is located near the putative endonuclease domain (Appendix A), which we hypothesized might potentially impact the functioning of this domain. The second change found is a poly T region located between nt positions 1172 and 1185 (Appendix A) encoding FFFSF, which in all other isolates is normally a sequence that encodes IAKTV (Appendix A). The domain structure of the RdRp was previously shown to consist of an N-terminal endonuclease domain and polymerase motifs (preA, A, B, C, D, and E). The locations of these two mutations were determined to be adjacent to the endonuclease domain in the linear diagram of the viral RdRp (Figure 3A). To better understand the location of the FFFSF mutation within the folded RdRp structure, we compared the reference RRV RdRp sequence (NCBI RefSeq: NC_105298) and a modified RRV RdRp containing the conserved IAKTV segment with a canonical bunyavirus RdRp structure as a reference (Appendix A). The I-TASSER server was used to generate three-dimensional structures using the LaCrosse virus (LACV) RdRp as the threading template (Appendix A, Appendix A) [33,34,35]. Looking first at the polymerase domain, the mutated segment containing the FFFSF sequence appears to lie in a region analogous to the LACV RdRp, and this region is near the viral RNA (vRNA) binding loop (vRBL). The LACV vRBL is an alpha-helical lobe that is mainly involved in interactions with the vRNA promoter (Appendix A). The mutated region’s proximity to the vRNA promoter suggested it was reasonable to consider that the changes in the NCBI reference sequence may impact viral RNA replication. Deleting the A near the 5′ end of the alignment eliminated the frameshift and added a helical arm to the end of the endonuclease domain (Appendix A).

Next, the N coding and protein sequences were aligned against recently deposited sequences in NCBI GenBank using MUSCLE. Substitution changes in the reference sequence (NC_015300) were located between nts 640 and 651 that were not found in any other sequences (Appendix A). This substitution resulted in a change of four amino acids encoding EFAL in the reference protein sequence (YP_004327589), whereas most N proteins encode NISE. The three-dimensional structures obtained using the I-TASSER server did not produce robust threading templates for making inferences about the impacts of these specific changes (data not shown).

We introduced mutations into the RRV R1 and R3 constructs to resemble the consensus sequences seen in most RRV isolates. The construct R1delA had the extra adenine deleted, R1subT had a substitution of the poly(T) stretch so that the conserved elements encode IAKTV, and R1delAsubT had both changes. The construct R3NISE had the coding sequence for EFAL replaced with the coding sequence for NISE (Figure 3A).

*Agrobacterium* cultures with these modified constructs were co-infiltrated with R5GFP plus the silencing suppressor p19 into *N. benthamiana* leaves at a concentration of 0.3 OD_600_ (Figure 3B). The mixtures were R1 plus R3, R1delA plus R3, R1subT plus R3, R1 plus R3NISE, R1delAsubT plus R3, and R1delAsubT plus R3NISE. We included as controls R5GFP alone plus p19, and the empty plasmid pCB301. Using a platform-fixed UV lamp, the infiltrated regions could be seen fluorescing at 3 dpi for all treatments except the controls. Fluorescence mostly faded at 5 dpi across most treatments, but interestingly, R1delAsubT plus R3NISE still had some visible fluorescence at 5 dpi. This suggests that the mutations that were introduced to the constructs had a positive influence on GFP expression over time (Figure 3B,C).

Fluorometric analysis of samples were performed at 3 and 5 dpi for each treatment and the average values (FV/mg-tissue) were determined (Figure 3D). The experiment was repeated three times and statistically analyzed as before. The average values were similar across all treatments at both time points (*n* = 6, *p* < 0.10). However, it is noteworthy that treatment with R1delA plus R3 produced the highest average fluorescence at 3 dpi with an average of 5236 FV/mg-tissue. All other treatments at the same time point had averages that ranged from 3054 to 3379 FV/mg-tissue. A second notable difference was at 5 dpi where the combination of R1delAsubT plus R3NISE caused fluorescence to rise from an average of 3379 FV/mg at 3 dpi to 3619 FV/mg at 5 dpi. This was significant because it was the first situation where GFP fluorescence was stable between 3 and 5 dpi. Together, these data suggest that the combined changes in R1 and R3, which resemble the majority of RRV isolate sequences, are important for prolonging GFP expression and for potentially prolonging virus replication (Figure 3D).

### 3.4. Detection of Plus and Minus-Strand Virus RNA Accumulation Using Tagged Primers

Genome replication for RRV, as for all bunyaviruses, occurs via a complementary positive-stranded RNA [10,36]. To examine the potential replicative amplification of the R5GFP template, we established endpoint and quantitative reverse transcription PCR (RT-PCR and RT-qPCR) assays to examine R5GFP positive and negative sense RNAs produced during virus replication. To protect against non-specific cDNA amplification which has been reported to occur when performing in vitro reverse transcription reactions using standard reverse transcriptase enzymes at 42 °C [37], we used a reverse transcriptase enzyme with RNAse H activity to degrade RNA and the reactions were performed at 65 °C to reduce self-priming that may be caused by RNA secondary structure [38,39]. Tagged primers are recommended for strand-specific reverse transcription PCR to reduce or eliminate detection of false-primed cDNAs, and to increase the accuracy of strand-specific quantification of viral RNAs [37,40,41,42,43]. Therefore we used the tagged primers GFP209-RT and GFP414-RT (Appendix A and Figure 4A) [37,44,45] which contain 20 unique nts (TAG) at the 5′ ends such that the first round cDNA also incorporates this 5′ tag (Appendix A). Then we performed PCR using the gene-specific primers GFP209F or GFP414R, along with a 20 nt TAG primer (Appendix A and Figure 4A).

To validate this approach, the GFP coding sequence was cloned into the plasmid pGEM-T (pGEMT-GFP) (Figure 4A) and T7 and SP6 polymerase were used to produce sense (ag) and antisense (g) GFP transcripts. Transcripts were diluted 10-fold from 1.0 ng to 0.1 pg and used to create cDNA in both orientations with the GFP414-RT or GFP209-RT primers (Appendix A) [43]. We included one control (C1 and C3) that lacks the reverse transcriptase and another control (C2 and C4) that uses in vitro synthesized transcripts of the same orientation as the primer used for the reverse transcription. These controls demonstrated that the PCR products derive from specific cDNA templates and were not the result of false priming (Figure 4B). Then PCR was performed using the respective cDNA templates along with the GFP209F, GFP414R, and TAG primers. We detected as little as 1.0 pg of gRNA and 100 pg of agRNA. There was no PCR amplification in the control reactions (Figure 4B).

Next, we agroinfiltrated leaves to deliver the original R1+R3+R5GFP+p19 minireplicon, the modified R1delA or R1subT+R3+R5GFP+p19, R1+R3NISE+R5GFP+p19, R1delAsubT+R3+R5GFP+p19, and the R1delAsubT+R3NISE+R5GFP+p19 minireplicons (Figure 4C). Controls included R5GFP+p19 or the pCB301 empty vector. Extracted RNA was treated with DNAse I to remove potential contaminating binary plasmids. Then endpoint RT-PCRs were performed using the tagged GFP414-RT or GFP209-RT primers. For controls, we did not include reverse transcriptase or the first strand primer to assess the potential for amplifying contaminating plasmid DNA or for false priming during first-strand synthesis, respectively (Figure 4C). We selected actin as an endogenous control to ensure equal loading on an ethidium-stained agarose gel. When we excluded the first strand primer (GFP209-RT or GFP414-RT) or enzyme from the reaction, we did not see bands (Figure 4C), suggesting that these reaction conditions were devoid of the potential for primer independent cDNA synthesis and the potential amplification of plasmids that might contaminate the RNA extracts. However, a band occurred in the gRNA sense R5GFP in the sample infiltrated with only R5GFP plus p19 (Figure 4C, lane 19), suggesting that this endpoint assay does not accurately detect the correct strand. The PCR bands generated using the original R1 plus R3 (Figure 4C, lanes 1, 2, and 3) and the negative control R5GFP plus p19 by themselves were similar (lanes 19, 20, and 21). Given that R5GFP plus p19 should produce only transcripts and not gRNA or replicating intermediates, there may be unavoidable carryover of the first strand primers from the cDNA synthesis step into the PCR. However, with equal sample loading on the gel, we noticed stronger bands in all other samples relative to R5GFP+p19, most notably for the R1delA+R3+R5GFP+p19 minireplicon (lane 4), or R1delAsubT plus R3NISE+R5GFP+p19 (lane 16) samples. The PCR products representing agRNA for R1subT or R1delAsubT+ R3+R5GFP+p19 (lanes 7 and 10) produced slightly stronger bands than the gRNA.

Several attempts to improve these RT-PCR results by using exonuclease I to eliminate potential GFP209-RT and GFP414-RT primers that might carry over into the PCR failed (data not shown). However, we found that reducing the GFP209-RT and GFP414-RT primers from 100 nM to a final concentration of 10 nM in the first step of cDNA synthesis resulted in higher levels of PCR products from replicon samples relative to the R5GFP+p19 control. Therefore, we performed RT-qPCR to measure the accumulation of viral gRNA and agRNA relative to any non-specific amplification obtained with the R5GFP+ p19 control. Before RT-qPCR involving the minireplicon, we optimized the primer concentrations for RT-qPCR using 100 pg/µL in vitro-transcribed GFP RNA spiked in 10 ng of healthy RNA from uninfected *N. benthamiana*. Melting curve assays showed a single amplification peak representing a single RNA species (Appendix A).

For all minireplicon samples, the RT-qPCR results show that the levels of gRNA and agRNA at 3-dpi were significantly greater compared to samples treated with only R5GFP, and no amplification was observed in samples treated with only pCB301 (Figure 4D). Consistent with the earlier results of fluorometry experiments, the gRNA and agRNA produced by the original R1+R3+R5GFP+p19 was an average of 4.8- and 4.5-fold, respectively, above R5GFP alone (average of 1.0) at 3 dpi. However, these levels declined to 1.5- and 1.0-fold, at 5 dpi. At 3 dpi for R1delA+R3+R5GFP+p19, the levels of gRNA and agRNA were 3.3- and 7.7-fold, respectively, indicating that the delA mutation improved the agRNA accumulation specifically. At 5 dpi the levels for gRNA and agRNA were 1.8- and 1.3-fold, indicating that this mutation did not produce sustained increases in gRNA or agRNA accumulation. At 3 dpi for R1subT+R3+R5GFP+p19, the levels of gRNA and agRNA were 1.1-fold and 3.5-fold, respectively. At 5 dpi the levels of gRNA and agRNA were 3.0-fold which was higher than the original replicon at 5 dpi, suggesting that this mutation resulted in the delayed accumulation of gRNA from 3 to 5 dpi while no delay was seen for agRNA accumulation. The levels of agRNA were similar between 3 and 5 dpi. At 3 dpi for R1delAsubT+R3+R5GFP+p19, the levels of gRNA and agRNA were 8.5- and 8.9-fold, respectively. At 5 dpi the levels of gRNA and agRNA were 1.7 and 2.6-fold, respectively. The combined delAsubT mutations in RNA1 led to higher gRNA and agRNA levels than the original replicon at 3 but not 5 dpi (Figure 4D).

Next, we introduced the NISE mutation into R3 within the replicon and co-delivered that with the original R1 or the double mutant R1delsubT. At 3 dpi, the levels of gRNA and agRNA for R1+R3NISE+R5GFP+p19 were 2.6- and 2.0-fold, respectively. At 5 dpi the levels of gRNA and agRNA were 2.1 and 1.3- fold, respectively. These values were like the original replicon suggesting that the NISE mutation by itself hampers viral replication. However, at 3-dpi for R1delAsubT+R3NISE+R5GFP+p19, the levels of gRNA and agRNA were 6.1- and 4.1-fold. At 5 dpi the levels of gRNA and agRNA were 4.0- and 4.6-fold, respectively. These were the highest values at 5 dpi indicating that the combined mutations in R1 and R3 were important for sustainable replication. In the next experiments, this triple mutant replicon will be referred to as the optimized RRV-GFP replicon (RRVop-GFP).

### 3.5. Contributions of Three Silencing Suppressor Proteins to RRVop-GFP Replicon

Given the value of incorporating silencing suppressor proteins into the replicon systems for several other reported NSR viruses of plants [17,19,20], we compared the use of three silencing suppressors alongside the RRVop-GFP. We co-delivered plasmids encoding the cucumber mosaic virus (CMV strain FNY) 2b, the TBSV p19, and the tomato golden mosaic virus (TGMV) AL2 alongside the RRVop-GFP replicon. Leaves were examined with a UV lamp at 3 and 5 dpi. We observed higher fluorescence when using 2b, AL2, p19, or all three silencing suppressors with the RRVop-GFP replicon than in leaves treated with R5GFP+p19 alone. In particular, the co-delivery of the RRVop-GFP with FNY-2b produced a robust fluorescent signal at 3 dpi which seemed to increase at 5 dpi (Figure 5A). Next, six leaves were infiltrated with each treatment and then extracts were analyzed fluorometrically at 3 and 5 dpi. The average fluorescence (FV/mg tissue) at each time point was again standardized by deducting the average FV/mg of the R5GFP +p19 control. The results of three experimental replicates were analyzed by ANOVA followed by Tukey’s HSD, and a representative experiment is displayed in Figure 5B. RRVop-GFP +p19 expressed 7438 FV/mg of tissue at 3 dpi and 9221 FV/mg at 5 dpi. At 3 and 5 dpi, RRVop-GFP +AL2 showed 9519 and 6692 FV/mg, respectively. At 3 dpi, RRVop-GFP +2b expressed 9520 FV/mg, but drastically increased to 21,442 FV/mg at 5 dpi. For each treatment, the GFP levels were significantly higher in replicon-expressing leaves than the control and mock treated leaves (*p* < 0.1 at 3 dpi and *p* < 0.04 at 5 dpi). The surprising increase of the RRVop-GFP plus 2b from less than 10,000 FV/mg at 3 dpi to greater than 20,000 FV/mg at 5 dpi suggested that FNY-2b has a particularly positive effect on promoting virus replication and GFP expression. This marks the first evidence of a drastic increase in GFP fluorescence between 3 and 5 dpi.

Next, strand-specific RT-qPCR was carried out to examine the levels of gRNA and agRNA accumulation. At 3 dpi for RRVop-GFP +p19, the levels of gRNA and agRNA were 5.1 and 14.0, respectively. At 5 dpi the levels of gRNA and agRNA were 2.8- and 2.6-fold, respectively (Figure 5C,D). Surprisingly, at 3 dpi for RRVop-GFP+AL2, the levels of gRNA and agRNA were 4.2- and 7.8, respectively. At 5 dpi the levels of gRNA and agRNA were 2.6- and 1.2-fold, respectively. It appears that AL2 had a remarkable and unexpected suppressive effect on RRV replication. At 3 dpi for RRVop-GFP+2b the levels of gRNA and agRNA were 6.5- and 12.2-fold, respectively. At 5 dpi the levels of gRNA and agRNA were 9.3- and 21.9-fold, respectively. When using FNY-2b, levels of gRNA and agRNA were in the same range as when we used p19 at 3 dpi, but the levels of agRNA were much higher at 5 dpi suggesting that FNY-2b has a specific positive role in positive strand RNA accumulation.

The consequences of using different silencing suppressors on RRVop-GFP replication and positive-strand RNA accumulation were intriguing. Therefore, we co-delivered all three suppressors with the RRVop-GFP to assess their combined effects on virus replication and were surprised that they did not have an additive effect on the levels of agRNA. At 3 dpi with RRVop-GFP+p19+AL2+2b, the levels of gRNA and agRNA were 1.7- and 2.9-fold. At 5 dpi the levels of gRNA and agRNA were 3.1- and 8.0-fold.

### 3.6. Virion Assembly by Providing the R2 Encoding Pre-GP alongside RRVop-GFP Replicon plus FNY-2b

To further demonstrate that the RRVop-GFP replicon produces a replicating virus, we co-delivered the R2-encoding cDNA with the RRVop-GFP +2b replicon system to *N. benthamiana* leaves. We introduced a 6x His-tag at the N terminus of the pre-GP for immunodetection. Between 3 and 5 dpi, leaves that were infiltrated with RRVop-GFP +2b and RRVop-GFP +R2His+2b showed robust GFP fluorescence (Figure 6A). Next, we performed immunosorbent electron microscopy using commercial His antisera to recover virions onto grids. Using transmission electron microscopy, we observed bright particles that were around 0.2 µm in diameter (Figure 6B). Next, we harvested leaf sap from RRVop-GFP +2b and RRVop-GFP +R2His+2b expressing leaves and used this to inoculate fresh *N. benthamiana* plants. The virions did not seem to cluster and were spherical or pleomorphic under negative staining. We then attempted to passage RRVop-GFP +R2His+2b from an infected *N. benthamiana* via sap transmission. *N. benthamiana* leaves inoculated with sap from an infected *N. benthamiana* were monitored for up to 5 dpi using a platform-fixed UV lamp and by epifluorescence microscopy, but we failed to detect evidence that infection was passaged to a healthy leaf (data not shown). These data suggest that the minigenome mimics viral genomic RNA in its encapsidation as well as transcription and replication. However, the lack of infectivity by passaging particles to a new host suggests that these are not fully reconstituted viruses. Additional activities are likely needed that are encoded by the missing genome segments to enable the infection to passage between hosts.

## 4. Discussion

This RRV-GFP minireplicon system is suitable for reverse genetic studies of virus replication, encapsidation, and the contributions of viral silencing suppressors in NSR virus infection in inoculated leaves. Steps to bolster virus replication and GFP expression included optimizing the concentration and delivery of *Agrobacterium* cultures, identifying key mutations in the viral RdRp and N that contributes to robust replication and gene expression, as well as identifying and employing the CMV FNY-2b silencing suppressor protein to effectively counter host defenses. Minireplicon systems have enormously benefited the current state of knowledge of NSR virus replication and encapsidation [24,46,47] by enabling investigations into essential biochemical features of the viral replicase, *cis*-acting elements in the viral RNA, and accessory viral and host factors [48]. Recent successes with plant infecting NSR viruses use a general approach to construction that includes the introduction of agRNA into a binary vector with the 35S promoter to drive transcription. The replicons for SYNV, BYSMV, and TSWV use self-cleaving hepatitis delta virus ribozyme (HDV) and a hammerhead ribozyme (HH) to cleave transcripts near the 5′ and 3′ termini to drive the production of exact or nearly exact ends of the viral genomic RNA, but the RRV system requires only a single HDV ribozyme close to the 3′ end of the agRNA [16,19,20]. While the minireplicon systems for many positive-strand RNA viruses comprise a defective-interfering RNA, satellite RNA, or modified genomic RNA segment that contains a reporter gene such as GFP surrounded by the appropriate *cis*-elements required to drive replication and transcription, we used the RNA5 segment to engineer a reporter template for studying virus replication [49,50,51,52,53].

To optimize GFP fluorescence levels, we examined the delivery of different concentrations and proportions of R1 and R3 alongside OD_600_ of 1.0 for R5GFP and p19 and saw that reducing the OD_600_ concentrations from 1.0 to 0.3 increased fluorescence from ~3000 to ~5000 FV/mg at 3 dpi and from ~1000 to ~2000 FV/mg at 5 dpi. Based on these results we reduced the concentrations of all constructs to OD_600_ of 0.3 for all subsequent experiments, recognizing that additional improvements are needed to enhance and extend virus replication and GFP expression.

We identified three changes in the NCBI reference genome of RRV, which was the basis for the first reported infectious clone [25], that negatively impact virus replication. The bunyavirus RdRp and N are well studied for many species including crystallographic data [34,35,54,55,56,57] which enabled us to develop hypotheses concerning the impacts of these changes. Using the available structural models for related bunyavirus RdRp and N proteins, we selected the LACV RdRp as the closest model but failed to identify a robust model for the RRV N protein. Regarding the RRV RdRp, we identified an added adenine at position 53 which caused a frameshift and premature termination upstream of a secondary ATG start codon. In comparison with other RRV isolates, the predicted start codon lies at nt position 44, but in the reference genome and the initial RRV infectious clone [25], the translational start codon was predicted at nt position 108. Based on the sequence analyses, this likely caused an N-terminal 21 amino acid truncation of the RdRp. The second mutation was a unique stretch of thymines between nt position 1173 and 1185 and amino acid sequence FFFSF replacing the IAKTV found in most other RRV isolates. This mutation likely produced changes near the RNA binding region of the polymerase (Appendix A). The third mutation in the nucleocapsid protein between nt positions 640 to 651 encodes EFAL while all other RRV isolates encode NISE at that position. Given that the frameshift mutation, the FFFSF amino acid segment in the RdRp, as well as the NISE mutation in the N, were unique to only one reported RRV isolate, we suspected that there was selection pressure at these nt positions and amino acid residues that influence viral replication [58], which we confirmed through mutational analysis using the RRV-GFP replicon. Both GFP and RNA accumulation benefitted from the combined mutations at 3 and 5 dpi.

For many NSR viruses infecting animals, plants, and insects, the putative endonuclease domain is important for cap snatching activity as well as RNA replication [59,60,61,62]. The domain is connected to the rest of the protein by the linker domain which adjoins the putative palm domain [35,58]. The endonuclease cleaves the host capped RNAs and enables the short-capped RNA primers to serve as primers for viral mRNA transcription by interacting with the catalytic site of the RdRp for RNA synthesis. Rice stripe virus (RSV) is also capable of stealing capped RNA primers from other viruses, such as CMV [36,60,62,63,64]. The endonuclease domain is also important for processive elongation by the RdRp through the viral RNA template [35]. Combining the delA and subT mutations in R1 led to higher levels of gRNA and agRNA at 3 dpi but not 5 dpi, supporting the hypothesis that these mutations were important for mRNA synthesis and establishing virus replication. However, virus replication appeared to decline at 5 dpi. Combining R1delAsubT and R3NISE led to sustained GFP levels as well as gRNA and agRNA at 5 dpi, indicating that the N and the NISE mutation act in concert with the RdRp for processive elongation.

The N may stabilize the RNA or the RdRp during RNA synthesis, although further experiments are needed to understand the emaravirus RNA synthesis machinery. Reverse genetic studies of BUNV N demonstrated its influence in RdRp recognition of RNA templates in addition to forming RNP complexes for encapsidation [57,65,66]. For hantavirus, the N has RNA chaperone activity facilitating the formation as well as dissociation of RNA panhandles during replication [67,68,69]. For HTNV and TSWV, the N protein accumulates in cytoplasmic processing bodies (PB) and shows high affinity for the 5′ cap of host mRNAs and contributes to the viral cap-snatching mechanism [70,71,72]. Substitution mutations introducing NISE into the RRV replicon, when combined with changes in the R1 segment, sustained GFP expression at 5 dpi where otherwise it typically declines. These data indicate that the NISE mutation benefits the prolonged translation of the R5GFP template and requires optimizing changes in the RdRp RNA binding loop and the extension of the endonuclease domain. Future work is needed to dissect the amino acids contributing to interactions between the RdRp and N to understand how the delA, subT, and NISE changes influence RNA stability, RNA replication, cap-snatching, and/or viral RNA translation. Given that the RdRp and N are multidomain and multifunctional proteins, it can be difficult to conduct extensive mutagenesis into the structure and function without a viable 3D model. While the threading model cannot replace a true 3D crystal structure, the iTASSER approach provided sufficient structural information to make predictions that could be matched by experimental data [56]. We attempted alignment using default settings and did not find sufficient conservation at the amino acid level, and we also used the iTASSER server to compare the N with structures in the PDB, however, the output models were not statistically supported (data not shown). Future research would benefit from studies to resolve the RRV N structure to understand N–N protein interactions, N–RNA interactions, and N–RdRp interactions.

Evidence of NSR virus replication requires detection of the minus strand genomes, which has been challenging for studies involving RRV [14,25]. The genomic RNAs for RRV tend to be in very low abundance and may be stabilized in ribonucleoprotein complexes. For many NSR viruses, researchers have adopted strand-specific RT-PCR assays but have acknowledged the difficulty in accurately detecting the correct strand [38,40,43]. To overcome these challenges, we used synthetic transcripts to optimize the RT-PCR conditions, DNase I to remove template DNA from extracted RNA, and exonuclease I to remove unincorporated RT primers from the cDNA that may contaminate the PCR amplification [43,73]. There are also reports, such as for influenza A virus, of primer-independent reverse transcription where there is significant RNA secondary structure that can lend itself to priming reverse transcription, especially when reverse transcription is performed at 42 °C [39,45,74]. Researchers reported that raising the reaction temperature improved the specificity of the reverse transcription, especially when working with virus-infected samples where both the minus- and positive-strand RNAs are present [37,44]. Researchers often report using nested or tagged RT-PCR primers to prevent false priming when assessing virus replication. Other studies, such as with RSV or RVFV, reported that using tagged strand-specific primers for first strand synthesis followed by qPCR in a two-step reaction provided the ability to report the selective amplification of gRNA and agRNAs [39,40,75]. In this study, we adopted the tagged-primer approach for endpoint RT-PCR and RT-qPCR assays. In using synthetic transcripts to determine the accuracy and sensitivity of detecting GFP sequences on both minus and positive-strand RNAs, we learned that detection of gRNA was more sensitive than agRNA using the RRV primers designed for this study (Figure 4B). When using total RNA extracted from agro-infiltrated leaves as the basis for RT-PCR, removing any contaminating binary plasmids, reducing the concentration of reverse primer used for first strand synthesis, and optimizing the reaction temperature to avoid primer-independent cDNA synthesis were essential. Control assays were performed where the cDNA primer or the reverse transcriptase enzyme was excluded, and these showed no contaminating bands. However, in repeated endpoint RT-PCR assays there was an amplification of both R5GFP strands in the absence of the viral replicase components indicating that this approach has limited successes (Figure 4C). Using RT-qPCR assays we were able to report measurable differences between the gRNA and agRNA levels in RRVop-GFP expressing samples relative to the samples with R5-GFP alone.

This study employed heterologous silencing suppressors and not the RRV encoded silencing suppressors because until now these have not been characterized. Such work is lengthy and is likely to occur in the next study using the minireplicon. Emaraviruses are unusual because the species vary between 4 and 8 genome segments. RRV RNA5 and RNA6 were first reported in 2016 and researchers were uncertain whether they were genome segments or satellites [76]. Phylogenetic studies indicated such significant heterogeneity among the RNA5, 6, and 7 segments across Emaraviruses that deeper investigations into the functions of various RRV proteins are necessary to identify potential silencing suppressor proteins [6,10]. When testing the use of heterologous silencing suppressors to enhance virus replication and GFP expression, we found that the FNY-2b was superior to AL2 and p19 for enhancing GFP, gRNA, and agRNA levels, especially at 5 dpi. Other studies of BYSMV, RSV, SYNV, and TSWV use three silencing suppressor proteins in addition to the viral endogenous suppressor, including the potyvirus HC-Pro, the TBSV p19, and the hordeivirus gamma protein, but none tested the FNY-2b suppressor [16,17,18,19,20]. The FNY-2b is one of the first described silencing suppressor proteins. Similar to the p19 and HC-Pro, FNY-2b affects systemic silencing. FNY-2b binds siRNAs, as well as AGO1 and AGO4 in the cytosol and nucleus. FNY-2b interferes with DCL-2, -3, and -4 generated siRNAs that are 21, 22, and 24 nt in length [26,77,78]. Lewsey et al. (2010) showed that the FNY-2b counters jasmonic acid (JA) responsive gene expression while inhibiting the expression of some SA-responsive genes although enhancing other SA-regulated genes [79]. While JA is not considered a contributor to positive strand RNA virus resistance, new studies involving RSV, another segmented NSR virus, link viral JA signaling to antiviral defenses and identify viral transcriptional repressors that act to modulate JA signaling [80,81,82]. It is reasonable to consider the possibility that the antiviral role of JA may broadly suppress NSR viruses. These unexpected results suggest that these advances in NSR virus minireplicons for RRV and RSV present new opportunities for investigating the role of JA in NSR virus defenses.

The TBSV p19 interferes with AGO1 and AGO2 by sequestering double stranded siRNAs which have a specific role in antiviral silencing [83,84,85]. In this study p19 enhances replication and GFP expression from the minireplicon at 3 dpi, although both decline around 5 dpi. The TGMV AL2 inactivates adenosine kinase and blocks the methyl cycle that generates S-adenosyl methionine which is important for cytosine methylation in viral DNA and cellular histone methylation [27,84]. AL2 also interacts with AGO4, a component of RNA-directed DNA methylation, to prevent viral DNA methylation. AL2 induces a calmodulin-like protein Nb-rgsCaM which interacts with the turnip mosaic virus HC-Pro and tomato aspermy virus or the CMV Y strain 2b proteins to hamper silencing suppressor activities [86,87,88]. This latter interaction with Nb-rgsCaM can provide evidence for speculating about why combining all three silencing suppressor proteins with the RRVop-GFP mini-replicon in Figure 5 appears to suppress GFP levels at 3 dpi and hampers RNA levels at 3 and 5 dpi compared to using the FNY-2b suppressor. Importantly, co-delivery of the three silencing suppressors produced alternate levels of viral gRNA and agRNA when compared to using the individual silencing suppressors, suggesting that the activities of AL2 might interfere with p19 functions as well. These data argue that the FNY-2b is sufficient to counter host defenses and promote RRVop-GFP replication.

The ability to rescue virions by introducing the plasmid encoding agRNA2 to the replicon system was an exciting advance. Most bunyaviruses encode three segments of negative polarity that are packaged by the pre-GP. Given that RRV has seven genome segments, we were uncertain whether these four components were sufficient for virion packaging. Given the success in producing particles in Figure 6, we prepared sap extract from GFP fluorescing tissue but failed in attempts to mechanically disseminate infection to a new host (data now shown). These data indicate that virion packaging is not the minimum requirement for transmission, and we expect that including additional genome segments that encode essential transmission features will enable virus spread. This advanced RRVop-GFP replicon system creates a new opportunity to investigate the requirements for genome packaging. For example, in RVFV genome packaging experiments reported by Wichgers Schreur et al. (2014), the pre-GP gene was divided into individual genes encoding the Gn or Gc and this was shown to also reconstitute virions [89,90,91,92]. Such studies highlight the value of a replicon system for reverse genetic studies to explore the functions of the precursor glycoprotein and to obtain important new insights into the emaravirus packaging and provide the selective and exclusive features for recognition of genome segments. Very little is known about the structural or functional characteristics of viral envelope glycoproteins in plants and how they engage with plant membrane systems. This RRVop-GFP system can also be used in the future to investigate glycoprotein assembly on cellular membranes and the budding of mature virions in infected cells.

## Figures and Tables

**Figure 1 viruses-14-00836-f001:**
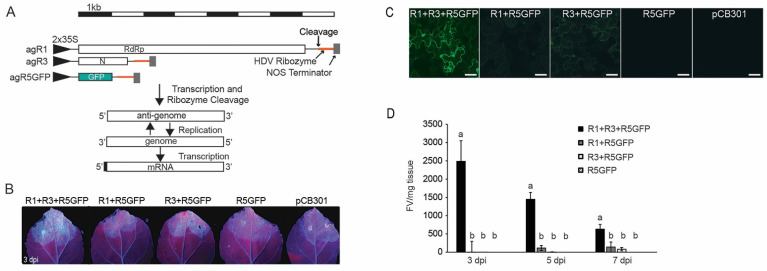
RRV-GFP replicon in *N. benthamiana* leaves. (**A**) The top is a scale bar divided into sectors representing 1 kb. Next is a diagrammatic representation of the pCB301-agR1, agR3, and -agR5GFP constructs. The black arrowheads represent the duplicated CaMV 35S promoters and the lines represent non-coding regions. Open boxes represent each ORF. The red bars identify the HDV ribozyme. The dark gray boxes represent the Nos terminator. The agR1 encodes the RdRp, agR3 encodes the N, and agR5GFP encodes GFP. Below the construct diagrams is a depiction of transcripts produced from the binary plasmids and subsequent replication of gRNAs and agRNAs. (**B**) Images of *N. benthamiana* leaves under a UV light at 3-dpi following agro-infiltration. Each panel label identifies the treatments. (**C**) Confocal microscopy image of *N. benthamiana* epidermal cells at 3-dpi. Scale bars = 20 µm. (**D**) Bar graph displaying the average FV/mg-tissue at 3, 5, and 7 dpi obtained from one representative experiment (*n* = 12). The letters above each standard error represent the statistical differences across treatments at each time point using Tukey’s HSD test with *p* < 0.1 Bars with the same letter are not significantly different for each time point.

**Figure 2 viruses-14-00836-f002:**
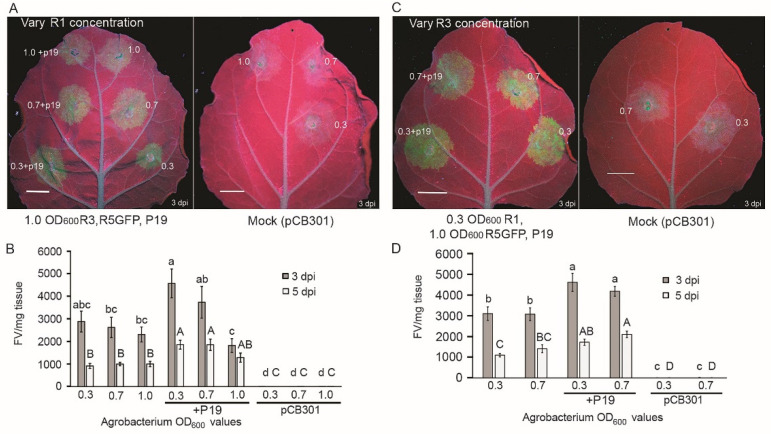
Optimizing *Agrobacterium* concentrations for delivery of RRV constructs. (**A**) *A. tumefaciens* harboring R1 were concentrated to 1.0, 0.7, and 0.3 OD_600_ and co-delivered with R3, R5-GFP, and p19 at concentrations of 1.0 OD_600_. Images show GFP fluorescence under a handheld UV lamp at 3 dpi. Scale bars represent 1 cm. (**B**) Bar graph displaying the average FV/mg tissue (*n* = 6) at 3 and 5 dpi obtained in one representative experiment. The lines through the bars represent standard error of the mean. The concentrations of *A. tumefaciens* cultures delivering R1 were 1.0, 0.7, and 0.3 OD_600_. Multiple comparisons of the means were performed using Tukey’s HSD (*p* < 0.05) and statistical relatedness is represented by letters next to each standard error bar. Values with the lowercase letter represent statistical analysis at 3 dpi and uppercase letter represent statistical analysis at 5 dpi. Bars with the same letters at 3 dpi or 5 dpi are not statistically different. (**C**) *A. tumefaciens* harboring agRNA3 were concentrated to 0.7 and 0.3 OD_600_ and co-delivered with 0.3 OD_600_ of agRNA1 and 1.0 OD_600_ of R5-GFP and p19. (**D**) Bar graph displaying the average FV/mg fresh weight tissue (*n* = 6) at 3 and 5 dpi obtained in one representative experiment using R3 at 0.7 and 0.3 OD_600._ The lower-case letters at 3 dpi and the uppercase letters at 5 dpi indicate the significance levels obtained using Tukey’s HSD (*p* < 0.05), and lines through the bars represent standard error of the mean. Values with the same letter are not statistically different.

**Figure 3 viruses-14-00836-f003:**
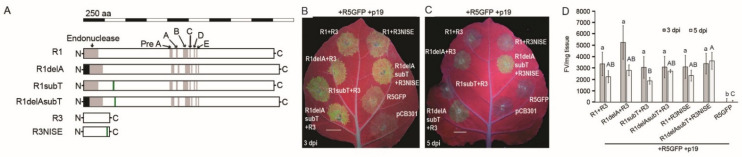
Mutagenesis analysis of RRV-GFP minireplicon. (**A**) Diagram of protein sequences and the location of the introduced mutations. The scale bar shows sectors representing 250 amino acids (aa). The name of the mutated protein is to the left. The gray boxes point to putative domains. The black boxes highlight the 21 amino acid extensions. The green boxes feature the poly T region in the nt sequence and the amino acid sequences FFFSF and IAKTV (R1subT), R1delAsubT has the deletion of adenine as well as substitution of the poly T region. Green boxes indicate another nt substitution to change the amino acid sequence from EFAL to NISE (R3NISE). (**B**,**C**) pCB301 constructs encoding the mutated RRV proteins in panel A were co-infiltrated into *N. benthamiana* in different combinations along with R5GFP +p19 and observed under UV light at 3 and 5 dpi. The white bars indicate 1 cm. The controls included R5GFP +p19 alone or the empty vector pCB301. (**D**) The infiltrated leaf patches were excised and used in fluorometry to quantify GFP fluorescence. The graph shows the average FV/mg- tissue and is representative of three experimental replicates (*n* = 6). The lines through the bars represent the standard error of the mean. Multiple comparisons of the means were performed by ANOVA and Tukey’s HSD (*p* < 0.05). Values with lowercase letters represent analysis across 3 dpi and uppercase letters represent 5 dpi. Mean values that are not statistically different are indicated by the same lowercase or uppercase letters.

**Figure 4 viruses-14-00836-f004:**
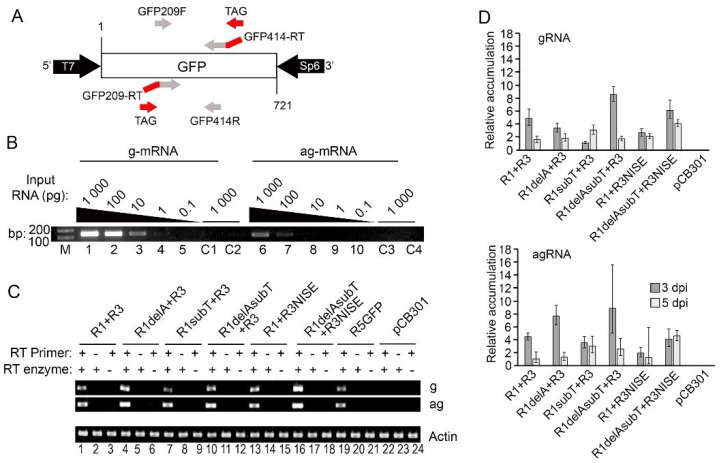
Strand-specific RT-PCR of a non-viral template to observe replication. (**A**) Diagram representing the pGEMT-eGFP construction and primers (arrows) for strand-specific RT-PCR amplification. Reverse transcription used GFP209-RT and GFP414-RT primers, each having a 20 nt tag sequence at the 5′ end (red). (**B**) Endpoint RT-PCR of GFP transcripts detecting gRNA (lanes 1–5, C1 and C4) and agRNA (lanes 6–10, C2 and C3). First-strand synthesis using serial dilutions (1000 to 0.1 pg) of transcripts are identified above lanes 1 to 5 and 6 to 10. C1 and C3 identify control reactions performed without the RT primer. C2 and C4 identify control reactions performed using the opposite RNA templates. All four control reactions used 1000 pg of transcripts. (**C**) Strand-specific endpoint RT-PCR detecting R5GFP RNA at 3 dpi following delivery of the wild-type and mutant RRV-GFP replicons. The control treatments were R5GFP +p19 only or the empty pB301 vector (lanes 19 to 24). Reactions performed with (+) or without (−) the primer or reverse transcriptase enzyme for first-strand synthesis are featured above the lanes. The bottom gel shows actin as internal controls. (**D**) RT-qPCR detecting GFP RNA in *N. benthamiana* leaves 3 and 5 dpi. The gRNA and agRNA levels are presented as relative to R5GFP +p19 using the ddCt method. The graph represents five experimental repetitions. The error bars highlight the ranges calculated from the mean Ct with 95% confidence intervals.

**Figure 5 viruses-14-00836-f005:**
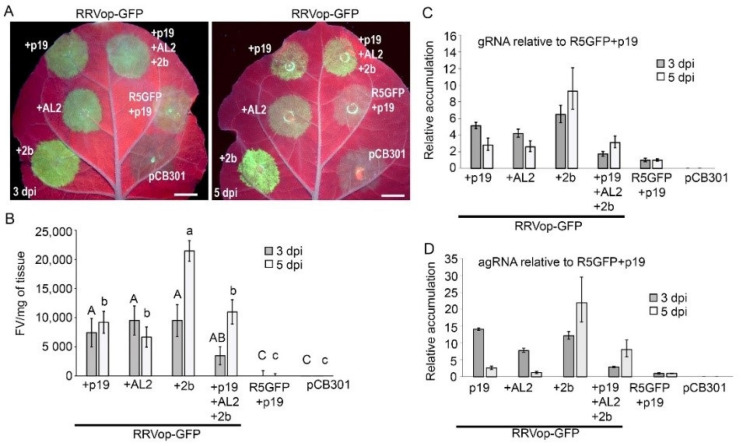
Co-delivery of three silencing suppressors with RRVop-GFP minireplicon. (**A**) Fluorescence images show the RRVop-GFP and silencing suppressors in *N. benthami*ana leaves at 3 and 5 dpi. Controls include the R5GFP +p19 only or the empty vector pCB301. Scale bars represent 1 cm. (**B**) The graph shows the average FV/mg- tissue and is representative of three experimental replicates (*n* = 6). Lines through the bars represent standard error of the means. The transcription levels of g (**C**) and ag (**D**) GFP RNA were measured via RT-qPCR using the tagged RT primer strategy. The results are presented as relative to R5GFP +p19 using the double delta Ct method. Actin primers amplified the endogenous control. The error bars represent ranges calculated from the mean Ct with a 95% confidence interval. For (**B**–**D**) multiple comparisons of the means were performed using ANOVA and Tukey’s HSD (*p* < 0.05). Lowercase letters indicate comparisons across 3 dpi and uppercase letters across 5 dpi. Values with the same lowercase or uppercase letter are not statistically different.

**Figure 6 viruses-14-00836-f006:**
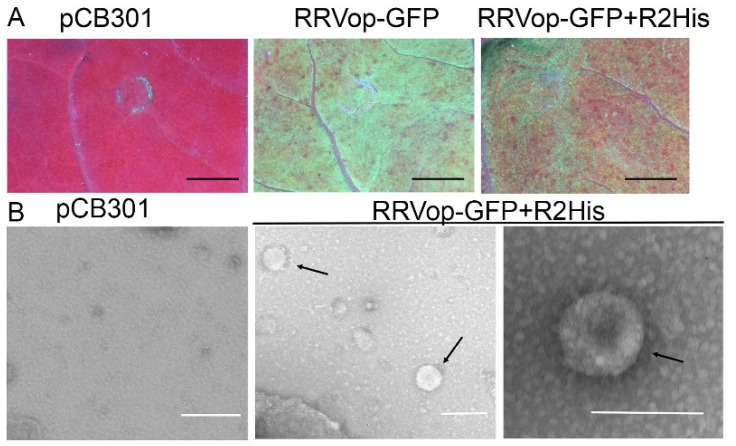
Adding agR2His to RRVop-GFP+2b and detection of virions in infected leaves. (**A**) Fluorescence imaging of *N. benthamiana* leaves treated with empty plasmid pCB301, RRVop-GFP replicon, or RRVop-GFP plus R2His. Black scale bars represent 1 cm. (**B**) Electron micrographs show mock pCB301 treated extract or virion particles (black arrows) in RRVop-GFP+R2His. White scale bars represent 0.2 microns.

## Data Availability

Data is made available through the Texas A&M University Libraries OAKTrust digital repository.

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
