# Peer review of "Advancing the Rose Rosette Virus Minireplicon and Encapsidation System by Incorporating GFP, Mutations, and the CMV 2b Silencing Suppressor"

_viruses, 2022, doi:10.3390/v14040836_

Round 1
Reviewer 1 Report
The manuscript authored by Urrutia et al. investigated the optimum Agrobacterium concentrations and proportions of R1 and R3 for delivery of RRV constructs using the minireplicon containing the GFP. Natural mutations delA and subT in the RdRp and NISE mutation in the N were introduced to clarify the influence of RNA synthesis and replication. The result also showed that the CMV 2b could singularly enhanced GFP expression and RRV replication. Specifically, by introducing a plasmid encoding agRNA2 to the replicon, the virions could be rescued but not to spread.
The manuscript is well written. The conclusion in this manuscript is interesting and valuable for the study of Emaravirus.
Therefore, I recommend ‘Accept with revision’.
Please see some specific points below:
Line 85, please use the abbr. pre-GP instead of precursor glycoprotein;
Line 89, Escherichia coli appeared in the title 2.1 but it is not mentioned in the following description;
Line 187, what is the concentration of the p19? Is it random or fixed? The same as below.
Author Response
We thank the reviewer for taking the time to carefully read the manuscript and offer useful suggestions. We used the abbr. pre-GP instead of precursor glycoprotein in line 85 as well as lines 473, 672, and 681.
We also added the name of the binary plasmid pBA002 for all silencing suppressors for clarification in line 94. line 187 we added 1.0 OD p19 for clarification.
We hope these are appropriate changes.
Reviewer 2 Report
The article entitled “Key mutations in rose rosette virus minireplicon and delivery of CMV 2b silencing suppressor enhance replication and encapsidation” is a straightforward article about the RRV MR system establishment and optimization. The manuscript is well written and fully covered all the aspects about RRV MR optimization. This innovation expands the molecular toolbox for investigating RRV replication and encapsidation. However, I feel the manuscript covered two independent story, and the title was only talked about the last one, Furthermore, the following suggestions should be carried out:
Abstract:
a concluding remarks section should be added at the end of the abstract to summarize the whole article.
Introduction
Line 65-66: replace “nonsegmented plant infecting NSR viruses” with “plant nonsegmented NSR viruses”
Line 66-67: replace “Sonchus yellow net virus (SYNV; a nucleorhabdovirus) and Barley yellow striate mosaic virus (BYSMV: a cytorhabdovirus)” with “Sonchus yellow net virus (SYNV) for nucleorhabdovirus and Barley yellow striate mosaic virus (BYSMV) for cytorhabdovirus”.
Line 68: replace “segmented” with “plant segmented NSR viruses,”
Line 69: “These existing minireplicon systems”, so this paragraph should about MR reverse genetic systems? As we know, TSWV has the infectious clone but RSV has only the MR.
Line 70-71: replace “viral antigenomic cDNAs” with “antigenomic-sense minireplicon RNA”
Line 151-157: the sentence should put in “Materials and Methods”.
Line 178-179: replace “improve” with “enhance”
Line 186: add “of R1” after in the “OD600”, same as line 195,196
Line 202-203: add “R3” in the back of “OD600”
Line 422: replace “ NSR viruses of plants” with “plant NSR viruses”
Line 819: the page and volume number should be added. And please read carefully about the whole reference section, like Line 870: “ / 835 MPMI” replaced with “MPMI” ,Line 898: “Advances in Virus Research” replaced to “Adv Virus Res”
Figure1D: Y axis better be changed in two segments to reflect the lower expression
Author Response
We thank the reviewer for their thoughtful reading and well thought comments for improving the manuscript. We adopted all the recommended changes. Notably we improved the title to reflect the work: "Advancing the rose rosette virus minireplicon and encapsidation system by incorporating GFP, mutations, and the CMV 2b silencing suppressor". We updated the first sentence and final concluding sentence of the abstract: "Plant infecting emaraviruses have segmented negative strand RNA genomes and little is known about the infection cycles due to the lack of molecular tools for reverse genetic studies" and then "In this study we developed a robust reverse genetic system for investigations into RRV replication and virion assembly that could be a model for other emaravirus species. "
We made the requested changes in the introduction. We moved lines 151-157 to the M&M section and agree that is a better location for it. We changed Improve for enhanced and went through all the OD values to make sure we properly referred to R1 and R3 and p19 as requested.
We updated the references one-by one checking each.
Finally regarding FIgure 1D. The values were so close to zero that we did not feel it warranted to split the Y-axis. These were barely above background as represented in this graph.
Reviewer 3 Report
Authors develped a rose rosette virus (RRV) minireplicon containing a GFP marker and analyzed amino acids in the viral the RdRp influencing the levels of gRNA, and agRNA. Authors also showed that RNA silencing suppressors from other viruses could enhanced RRV replication and observed virion-like particles. The study presented some new results for further understanding the protein functions and pathogenic mechnism of the virus by using reverse genetic systems.
The manuscript was well writtern and results were clearly presented.
Other commons:
It is better to confirm the roles of RNA silencing suppressors in the viral replication by providing northern blot results for the viral genome RNA and siRNA.
Authors failed to pass the virus particles basing on fluorescence observation. Why did not to do sensitive RT-PCR assays?
Author Response
We thank the reviewers for their thoughtful comments and reflections on this work. They suggested 2 experiments.
1) It is better to confirm the roles of RNA silencing suppressors in the viral replication by providing northern blot results for the viral genome RNA and siRNA. We are preparing a study of the various RNA silencing suppressors in our followup paper to dive into the various roles and contributions. The goal in this study was to enhance GFP expression which we achieved. The molecular mechanisms that led to various outcomes is the topic of our current study. So please stay tuned for more to come.
Authors failed to pass the virus particles basing on fluorescence observation. Why did not to do sensitive RT-PCR assays? I am not understanding the comment. We did not passage samples from one plant to the next. That is correct. Since we do not have a movement protein identified among the 7 segments we felt that we needed to expand the study to identify a movement protein so we can effectively passage virus. This system is restricted to studying replication and demonstrating virions form inside the infiltrated zones.
Reviewer 4 Report
This is a well written manuscript and was a pleasure to review. The experimental data produced are very convincing with meticulous detail given to all possible reasons for the results obtained. I could only find one minor issue on ln 297 in the Figure 3 legend where the authors refer to blue boxes in the figure. On my pdf copy of the manuscript, I could not discern any blue boxes but did find green ones in the R1subT and R1delAsubT as well as the green box in R3NISE.
Author Response
Thank you for reviewing the paper and we are glad for the appreciation of our work. We changed the statement from blue to green in the figure legend. Thank you for identifying that error.